# CTRL-WORLD: A CONTROLLABLE GENERATIVE WORLD MODEL FOR ROBOT MANIPULATION

**Yanjiang Guo**[*12]**, Lucy Xiaoyang Shi**[*1]**, Jianyu Chen**[2]**, Chelsea Finn**[1]
[*] Equal Contribution, [1] Stanford University, [2] Tsinghua University
Project page: `https://ctrl-world.github.io`

## ABSTRACT

Generalist robot policies can now perform a wide range of manipulation skills, but evaluating and improving their ability with unfamiliar objects and instructions remains a significant challenge. Rigorous evaluation requires a large number of real-world rollouts, while systematic improvement demands additional corrective data with expert labels. Both of these processes are slow, costly, and difficult to scale. World models offer a promising, scalable alternative by enabling policies to rollout within imagination space. However, a key challenge is building a controllable world model that can handle multi-step interactions with generalist robot policies. This requires a world model compatible with modern generalist policies by supporting multi-view prediction, fine-grained action control, and consistent long-horizon interactions, which is not achieved by previous works. In this paper, we make a step forward by introducing a controllable multi-view world model that can be used to evaluate and improve the instruction-following ability of generalist robot policies. Our model maintains long-horizon consistency with a pose-conditioned memory retrieval mechanism and achieves precise action control through frame-level action conditioning. Trained on the DROID dataset (95k trajectories, 564 scenes), our model generates spatially and temporally consistent trajectories under novel scenarios and new camera placements for over 20 seconds. We show that our method can accurately rank policy performance without real-world robot rollouts. Moreover, by synthesizing successful trajectories in imagination and using them for supervised fine-tuning, our approach can improve policy success by 44.7%.

## 1 INTRODUCTION

Recent advances in vision-language-action (VLA) models have demonstrated competence across a wide range of manipulation tasks and scenarios (Black et al., 2024; Wen et al., 2025; Brohan et al., 2023; Kim et al., 2024; Cui et al., 2025; Guo et al., 2025; Zhang et al., 2024). Despite their promise, current policies remain brittle when tested in open-world circumstances (Shi et al., 2025). A central challenge is *policy evaluation*. Assessing generalist policy performance typically requires large numbers of real-world rollouts, carefully repeated across tasks and environments to achieve statistical significance (Atreya et al., 2025). Such protocols are logistically demanding, slow down iteration, and inhibit nuanced understanding of current policy capabilities. Equally critical is *policy improvement*: once weaknesses are revealed, existing methods offer few ways to strengthen policies on failure cases beyond collecting more expert data. Although large-scale pretraining provides some robustness, policies often remain fragile when they encounter unfamiliar objects or instructions. What is missing is a fast and cheap feedback-driven mechanism for refining generalist models: a way to surface failure cases, gather corrective experiences, and iteratively improve the policy.

Learning a predictive model and iterating in imagination is a scalable and promising alternative. While prior work has explored action-conditioned world models, most approaches focus on passive video prediction settings and are not sufficient to actively interact with advanced generalist policies (Li et al., 2025b; Zhu et al., 2024). We observe several important limitations that hinder their ability to support policy-in-the-loop rollouts. First, these models typically simulate only a single third-person camera view, which can lead to severe partial observability and, in turn, cause hallucinations (e.g., an object snapping into the gripper without prior physical contact). This single-view input is also incompatible with many modern VLA policies that require both third-person and wrist-view cameras as input. Moreover, existing models typically lack the fine-grained control required to capture the

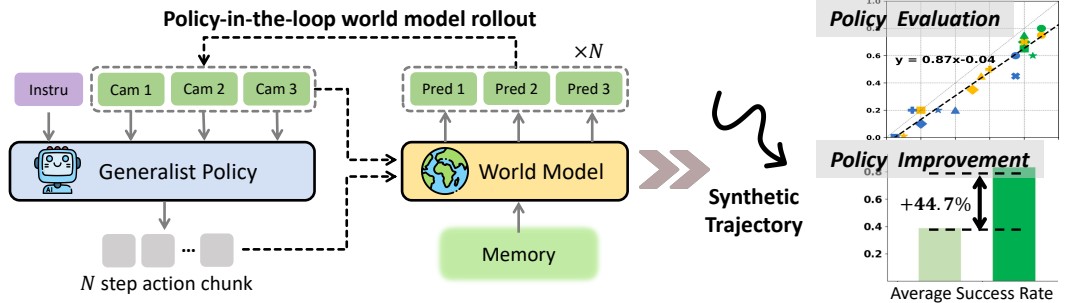

Figure 1: Ctrl-World is designed for *policy-in-the-loop* rollouts with generalist robot policies. It generates joint multi-view predictions (including wrist views), enforces fine-grained action control via frame-level conditioning, and sustains coherent long-horizon dynamics through pose-conditioned memory retrieval. These components enable (1) accurate policy *evaluation* in imagination, with alignment to real-world rollouts, and (2) targeted policy *improvement* through synthetic trajectories.

causal effects of high-frequency actions. Finally, they struggle to maintain temporal consistency across long-horizon video generations.

In this paper, we introduce **Ctrl-World**, a **Controllable**, multi-view generative **world** model designed for policy-in-the-loop interaction, enabling multi-step rollouts entirely within imagination space, as illustrated in Figure 1. Our design relies on three key components: (1) Joint multi-view prediction captures a more comprehensive visual representation of the scene and meets the input format of modern VLA policies. Notably, the inclusion of wrist-camera prediction significantly reduces hallucinations during contact-rich object interactions. (2) Frame-level action conditioning tightly aligns visual dynamics with control signals, ensuring that generated rollouts reflect the causal effect of each action. (3) Memory retrieval, which adds sparse history frames into the context and projects corresponding pose information into each frame, allows the model to attend to similar past states and retrieve relevant information. This mechanism stabilizes long-horizon rollouts and preserves temporal consistency. Together, these mechanisms allow us to transform a pre-trained passive video generator into a policy-compatible interactive simulator.

The core contribution of this work is a *controllable world model* for robot manipulation. In experiments, we find this model enables a new imagination-based workflow in which policies can be both *evaluated*—with ranking alignment to real-world rollouts—and *improved*—through targeted synthetic data that boosts success rates. Specifically, we train Ctrl-World on the DROID dataset (Khazatsky et al., 2024) and show that it generalizes to novel scenes and camera placements, sustaining coherent rollouts for over 20 seconds. We further show that imagination-based evaluations with Ctrl-World faithfully reflect policies' real-world instruction-following ability. Finally, we demonstrate that we can improve the performance of $\pi_{0.5}$-DROID (Intelligence et al., 2025) on downstream tasks with unseen objects and novel instructions by synthesizing successful trajectories inside the world model and performing supervised fine-tuning with these synthetic roll-outs.

## 2 RELATED WORKS

**Video Generation Models for Robotics.** Recent advances in video generation models (Agarwal et al., 2025; Wan et al., 2025; Blattmann et al., 2023a; Chi et al., 2025) have enabled the creation of realistic and temporally consistent content, reflecting a strong understanding of the physical world. Some works leverage video prediction models to synthesize robotic trajectories with fake action labels, and these synthetic trajectories can then be used for policy learning (Jang et al., 2025; Bharadhwaj et al., 2024). Other works directly employ video models as policy backbones, decoding actions through tracking or inverse dynamics (Black et al., 2023; Du et al., 2024; Yang et al., 2023; Hu et al., 2024; Liang et al., 2024; Liao et al., 2025; Tan et al., 2025; Feng et al., 2025). A complementary line of research integrates future-prediction objectives into generalist policies via co-training (Zhao et al., 2025; Li et al., 2025a; Zhu et al., 2025; Guo et al., 2024; Gao et al., 2024; Zhang et al., 2025; Zheng et al., 2025; Zhong et al., 2025), incorporating physical knowledge into the policy. Unlike these works, we leverage video generation to perform action-conditioned prediction, which enables using the model for both policy evaluation and policy improvement.

**Action-Conditioned World Models.** Although pretrained video models are powerful, they are often only conditioned on high-level language instructions. Nonetheless, some prior works have explored using action-conditioned predictive models, both in low-dimensional state spaces (Nagabandi et al., 2020) and with image observations Hafner et al. (2019; 2020); Hansen et al. (2022); Wu et al. (2023); Oh et al. (2015); Huang et al. (2025). Many of these approaches learn task-specific models (Hafner et al., 2019), while we focus on training generalist, multi-task world models. Building on early works (Finn & Levine, 2017; Ebert et al., 2018; Xie et al., 2019; Dasari et al., 2019; Yang et al., 2023; Wu et al., 2024) as well as more recent approaches that leverage diffusion (Quevedo et al., 2025; Chen et al., 2024; Ball et al., 2025; Gao et al., 2025; Ren et al., 2025; Hafner et al., 2025) and frame-level action conditioning (Zhu et al., 2024), we propose a model that incorporates multi-view prediction, long-horizon temporal coherence, and fine-grained controllability. Our experiments show that these capabilities enable effective evaluation and improvement of state-of-the-art generalist VLA policies.

## 3 PROBLEM FORMULATION

We aim to develop a world model that can predict the future outcomes of actions proposed by a generalist robot policy. A modern generalist policy $\pi$ typically maps multi-view observations and language instructions into a sequence of actions (Zhao et al., 2023; Black et al., 2025). Specifically, robot observation $o_t = [I_t^1, \ldots, I_t^n, q_t]$ includes $n$ camera views $[I_t^1, \ldots, I_t^n]$ and robot pose $q_t$, the policy outputs an $H$-step action chunk given an instruction $l$:

$$a_{t+1}, a_{t+2}, ..., a_{t+H} \sim \pi(\cdot|o_t, l) \tag{1}$$

Our goal is to use a world model $W$ to predict the outcomes of executing each step in $A_t = [a_{t+1}, \ldots, a_{t+H}]$. To enable multi-step interaction with the policy in imagination space, $W$ must generate future multi-view observations:

$$o_{t+1}, ..., o_{t+H} \sim W(\cdot|o_t, A_t) \tag{2}$$

Then the final prediction $o_{t+H}$ can be send back to policy $\pi$ to produce the next action chunk $A_{t+H} \sim \pi(\cdot|o_{t+H}, l)$. In this way, the policy and world model interact auto-regressively, enabling long-horizon rollouts entirely within imagination space.

## 4 CONTROLLABLE WORLD MODEL FOR ROBOT MANIPULATION

### 4.1 LEARNING WORLD MODEL CTRL-WORLD

Our goal is to learn a world model that can be used to evaluate and improve modern VLA policies. To achieve this, the model must first support multiview observations that are commonly used by such policies. It is also important for the model to be controllable — reliably and closely follow the action inputs — even when initialized from a pre-trained backbone that lacks such control. Finally, the model must maintain temporal consistency over long horizons, even in the presence of occlusions, to produce coherent rollouts. We initialize our world model from a pretrained video diffusion backbone with spatial-temporal transformers (Blattmann et al., 2023b) and introduce three key adaptations, illustrated in Figure 2.

**Multi-View Joint Predictions.** State-of-the-art VLA models often rely on multiple third-person cameras for global context and wrist-mounted cameras for precise interactions (Intelligence et al., 2025; Liu et al., 2024; 2025). To match this, the world model must generate spatially consistent predictions across all views at each step (Jiang et al., 2025). Prior work has shown that feed-forward transformers can effectively capture spatial relationships between multi-view cameras in a scalable manner (Wang et al., 2025). Following prior work, we concatenate the $N$ input images—each containing $H \times W$ tokens—along the token dimension and jointly predict all views $o_{t:t+H}$. In experiments, we find multi-view joint prediction also improves consistency and substantially reduces hallucinations.

**Pose-conditioned Memory Retrieval Mechanism.** Prediction errors in world models tend to accumulate over long rollouts, leading to drift and incoherence. To mitigate this, we augment the model input with past frames. To prevent the context from becoming too long, we sample $k$ history

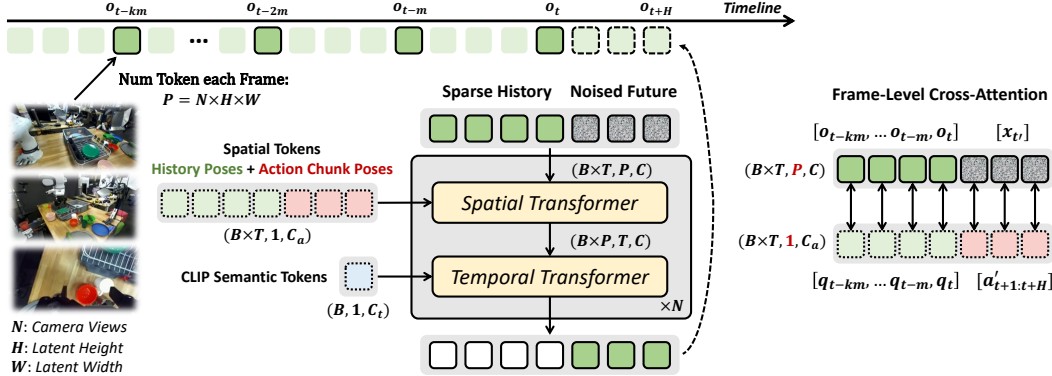

Figure 2: Ctrl-World is initialized from a pretrained video diffusion model and adapted into a controllable, temporally consistent world model with: (1) Multi-view input and joint prediction for unified information understanding. (2) Memory retrieval mechanism, which adds sparse history frames in context and project pose information into each frame via frame-level cross-attention, re-anchoring predictions to similar past states. (3) Frame-level action conditioning to better align high-frequency action with visual dynamics.

frames with a stride $m$, enabling the model to predict $o_{t+1:t+H} \sim W(\cdot | o_{t-km}, ..., o_t, l)$. Additionally, we embed the corresponding robot arm poses $[q_{t-km}, ..., q_t]$ into frames $[o_{t-km}, ..., o_t]$ via frame-wise cross-attention within spatial transformer. This allows the model to use the arm pose to identify relevant frames from the past, effectively anchoring future predictions to relevant history.

**Frame-level Action Conditioning.** The pretrained video model conditions only on text and image, which limits its control precision. To enable full controllability, we additionally condition the model on the action sequence $[a_{t+1:t+H}]$ output by the policy. We also transform each action sequence into Cartesian-space robot arm poses $[a'_{t+1:t+H}]$ and concatenate with past poses $[q_{t-km}, ..., q_{t-m}, q_t]$. Frame-wise cross-attention (Zhu et al., 2024; He et al., 2025) is then applied within the spatial transformer, allowing the visual tokens of each frame to attend to its associated pose embedding. For history frames, this pose corresponds to $[q_{t-km}, ..., q_{t-m}, q_t]$, while for future frames, it corresponds to $[a'_{t+1:t+H}]$.

**Training Objective.** We initialize our model with the pretrained 1.5B Stable-Video-Diffusion (SVD) model (Blattmann et al., 2023a). To inherit the knowledge and structure in the pretrained video model, we only newly initialize an action-projection MLP for the input actions and keep other parameters unchanged at initialization. Then this action-conditioned world model is fine-tuned with diffusion loss (Ho et al., 2020; Karras et al., 2022). During training, the prediction target $x_0 = o_{t+1:t+H}$ is perturbed with Gaussian noise $\epsilon \sim \mathcal{N}(0, I)$ at diffusion step $t' \in [0, T']$ with scheduler $\overline{\alpha_{t'}}$, resulting in $x_{t'} = \sqrt{\overline{\alpha_{t'}}}x_0 + \sqrt{1 - \overline{\alpha_{t'}}}\epsilon_{t'}$. The model input is the concatenation of history tokens and the noised future: $[o_{t-km}, ..., o_{t-m}, o_t, x_{t'}]$. The overall training objective is:

$$\mathcal{L} = \mathbb{E}_{x_0, \epsilon, t'} \|\hat{x}_0(x_{t'}, t', c) - x_0\|^2 \tag{3}$$

where $\hat{x}_0$ denotes the model's prediction, and $c = [q_{t-km}, ..., q_t, a'_{t+1:t+H}, o_{t-km}, ..., o_t]$ corresponds to all model inputs. More details of the model can be found in the Appendix A.

## 4.2 USING CTRL-WORLD FOR POLICY EVALUATION AND IMPROVEMENT

**Policy Evaluation within World Model.** Once a controllable and consistent world model is trained, we can conduct policy-in-the-loop rollouts in imagination space. Given an initial observation $o_0$ and instruction $l$, a policy $\pi$ together with the world model $W$ can generate a synthetic trajectory $\tau$. The initial observation can be sampled from the validation dataset or recorded as a snapshot from a real-world setup. In our experiments, we label each trajectory as a success or failure based on human preference judgments. While recent works (Du et al., 2023) explore the use of Vision-Language Models as general-purpose reward models, we leave such extensions to future work.

**Policy Improvement with Synthetic Data.** Beyond evaluation, the world model enables searching for successful synthetic trajectories to improve policy performance. We observe that, under fixed

---

**Algorithm 1 World Model Rollout and Policy Improvement**

---

**Given:** policy $\pi_\theta$, action perturbation function $\epsilon_a$, world model $W$, task instructions $[l^0, ., l^M]$ with initial obs $[o_0^0, ., o_0^M]$, synthetic dataset $D_s$, interaction step $N$, action horizon $H$.

 1: **for** $i = 0$ **to** $M$ **do**
 2:     $\tau = [o_0^i]$
 3:     **for** $j = 0$ **to** $N$ **do**
 4:         Current observation: $o_t = \tau[t]$ where $t = j * H$
 5:         Sample action from perturbed policy: $a_{t+1:t+H} = \pi_\theta(o_t, l, \epsilon_a)$     ▷ For diverse rollouts
 6:         Prepare history context: $h = [o_{t-km}, ..., o_{t-2m}, o_{t-m}]$
 7:         Make predictions with world model: $o_{t+1:t+H} = W(h, o_t, a_{t+1:t+H})$
 8:         Add predictions into trajectory: $\tau = \tau \cup o_{t+1:t+H}$.
 9:     **end for**
10:     Judge success of $\tau$ based on human-preference. Add $\tau$ into $D_s$ if success.
11: **end for**
12: Finetune $\pi_\theta$ with $\mathcal{L}_\theta = \mathbb{E}_{o_t, a_{t:t+H} \sim D_s} \|\pi_\theta(o_t, l) - a_{t:t+H}\|^2$.

---

initial observations and instructions, policy behavior tends to be highly deterministic. For example, the policy tends to grasp the same object across multiple trials, rather than stochastically reaching for various objects. To explore a larger search space, we introduce structured perturbations to encourage diversity in rollouts. Specifically, we can (i) rephrase the instructions, since VLA policies tend to be steerable, exhibiting different behaviors in response to different instructions; or (ii) reset the policy to random initial states within the world model, which leads to diverse initial observations. Starting from a set of downstream tasks with language instructions $[l^0, \ldots, l^M]$, we collect synthetic rollouts and score them based on human preference. To improve the policy performance, we fine-tune the policy on successful trajectories. The overall procedure is summarized in Algorithm 1.

## 5 EXPERIMENTS

In this section, we conduct experiments to evaluate Ctrl-World. We aim to answer the following questions: (1) Can Ctrl-World generate long-horizon rollouts that are both spatially and temporally consistent, while maintaining high controllability? (2) Can Ctrl-World reliably evaluate different generalist robot policies in imagination space, faithfully reproducing their real-world performance rankings? (3) Can Ctrl-World improve a policy's instruction following by discovering and synthesizing successful trajectories entirely within its imagination?

### 5.1 EXPERIMENT SETUPS

**DROID Platform and Dataset.** Our experiments use the DROID platform (Khazatsky et al., 2024), which features a Panda robot arm equipped with a Robotiq Gripper. The platform includes one wrist-view camera and two randomly positioned third-view cameras that observe the workspace. The DROID dataset (Khazatsky et al., 2024) contains 95,599 diverse trajectories collected from 564 scenes, providing dense coverage of the workspace. This includes about 76k successful and about 19k failed trajectories. The inclusion of diverse actions and failure data is crucial, as it allows us to train a controllable world model that can simulate a wide range of future scenarios.

**Training Details.** During training, our model jointly predicts outputs from all three cameras, each with a resolution of 192x320. The model is conditioned on a history of 7 frames, with an interval of 1-2 seconds between frames. We condition the model on the next 15 future actions, which corresponds to a one second action chunk in DROID. During interaction, if a policy's output is less than 15 steps, we pad the action chunk with dummy actions and only use the predictions for valid actions. We train the model on $2 \times 8$ H100 GPUs, with a total batch size of 64. Training takes approximately 2-3 days.

### 5.2 WORLD MODEL QUALITY ANALYSIS

**Baselines and Evaluation Matrices.** We quantitatively compare our model, Ctrl-World, against two prior action-conditioned world models: World-model-based Policy Evaluation (WPE) (Quevedo et al.,

| Evaluated Camera | Method | Computation-based | | Model-based | | |
|---|---|---|---|---|---|---|
| | | PSNR ↑ | SSIM ↑ | LPIPS ↓ | FID ↓ | FVD ↓ |
| Third-view Camera | WPE-Single-View | 20.33 | 0.772 | 0.131 | 25.50 | 156.4 |
| | WPE-Multiview | 21.17 | - | - | - | 147.1 |
| | IRASim-Single-View | 21.36 | 0.774 | 0.117 | 26.46 | 138.1 |
| | IRASim-Multiview | 20.21 | - | - | - | 165.4 |
| | Ctrl-World-Single-View | 21.27 | 0.793 | 0.110 | 23.47 | 127.5 |
| | Ctrl-World (ours) | 23.56 | 0.828 | 0.091 | 25.00 | 97.4 |

Table 1: Quantitative results for interactive long-trajectory generation on the validation set. We evaluate our world model's quality by generating 10-second trajectories. Given a randomly sampled initial frame, the model receives a 15-step action chunk (spanning over 1 second) in each interaction and generates for 10 rounds auto-regressively. The results are averaged over 256 clips.

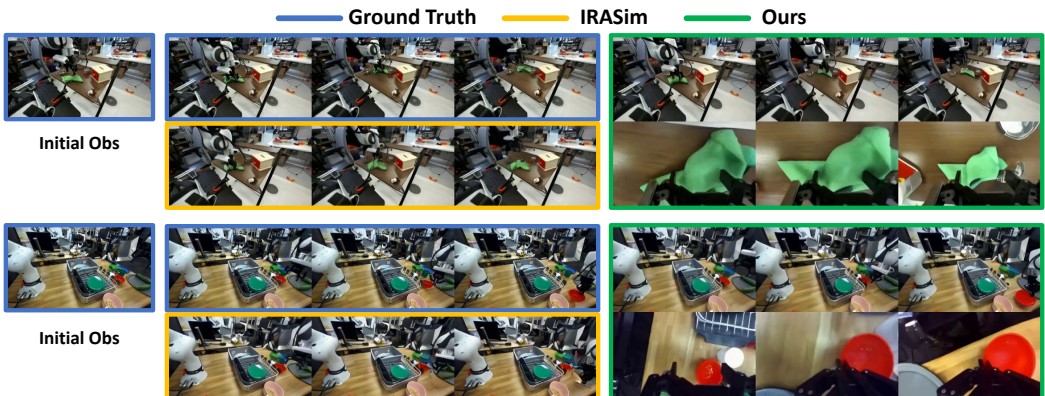

Figure 3: Qualitative results on long-horizon rollouts from the validation set. Prior models rely on single-view prediction, suffering from partial observability and hallucinations (e.g., failing to move the green towel or grasp the red bowl). In contrast, Ctrl-World jointly predicts from third-view and wrist-view cameras, yielding precise future trajectories aligned with the ground truth.

| Evaluated Camera | Method | Computation-based | | Model-based | | |
|---|---|---|---|---|---|---|
| | | PSNR ↑ | SSIM ↑ | LPIPS ↓ | FID ↓ | FVD ↓ |
| Third-view Camera | Ctrl-World | 23.56 | 0.828 | 0.091 | 25.00 | 97.4 |
| | Ctrl-World w/o memory | 23.06 | 0.812 | 0.099 | 26.14 | 105.5 |
| | Ctrl-World w/o frame-level cond | 21.20 | 0.789 | 0.109 | 27.52 | 122.7 |
| Wrist-view Camera | Ctrl-World | 19.18 | 0.665 | 0.252 | 25.78 | 127.1 |
| | Ctrl-World w/o memory | 18.84 | 0.655 | 0.265 | 26.23 | 133.1 |
| | Ctrl-World w/o frame-level cond | 15.69 | 0.571 | 0.375 | 33.51 | 179.1 |
| | Ctrl-World w/o joint pred | 15.94 | 0.580 | 0.345 | 26.46 | 158.1 |

Table 2: Ablations on key components in Ctrl-World. Removing memory mechanisms, frame-level action conditioning or multi-view joint predictions all lead to a performance drop.

2025) and IRASim (Zhu et al., 2024). Since these models only predict from a single third-person camera view, we train a single-view version, Ctrl-World-third-view, which only inputs and predicts on a single third-person camera for a fair comparison. For evaluation, we hold out 2% of the trajectories as a validation set and randomly sample 256 video clips, each 10 s in length. During rollouts, the world model receives 15-step action chunks (corresponding to 1 s) and autoregressively predicts the next frames for 10 steps, producing 10 s-long trajectories. We then compare the predicted videos against ground truth using both computational (PSNR (Hore & Ziou, 2010) and SSIM (Wang et al., 2004)) and model-based metrics (LPIPS (Zhang et al., 2018), FID (Heusel et al., 2017), and FVD (Unterthiner et al., 2018)).

**Quantitative and Qualitative Results on Multi-step Interaction Trajectories.** As shown in Table 1, Ctrl-World-third-view outperforms these prior models, and multi-view joint prediction further improves generation quality. Consistent with observations from prior work (Quevedo et al., 2025; Zhu et al., 2024), we also find that these baselines struggle to capture robot–object interactions and often generate hallucinated predictions. For instance, as shown in Figure 3, single-view prediction

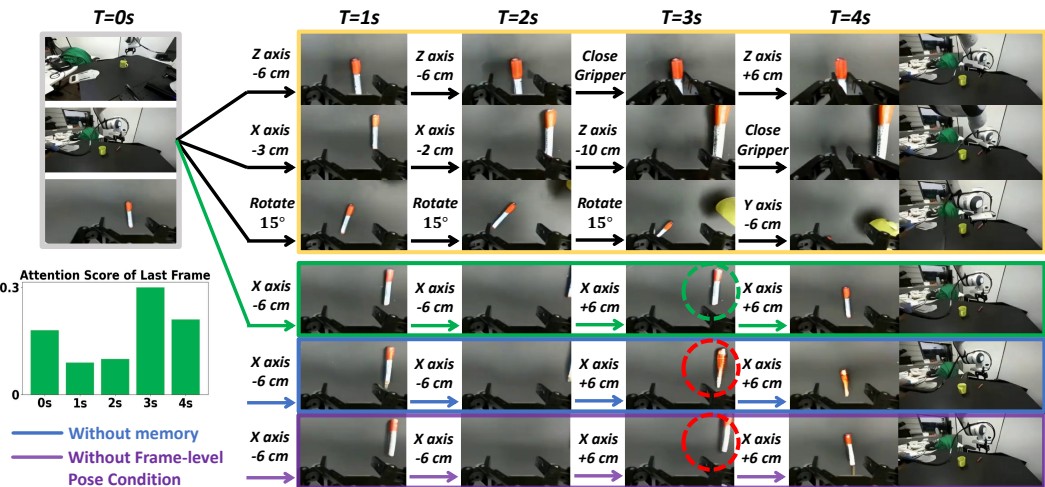

Figure 4: Controllability of Ctrl-World and ablations. Different action sequences can produce distinct rollouts in Ctrl-World with centimeter-level precision. Removing memory leads to blurry predictions (blue), while removing frame-level pose conditioning reduces control precision (purple). Attention visualization (left) when predicting the $t = 4\,\mathrm{s}$ frame shows strong attention to the $t = 0\,\mathrm{s}$ frame with the same pose, illustrating the effectiveness of memory retrieval. For clarity, each action chunk is expressed in natural language (e.g., "Z-axis -6 cm"). Due to space constraints, only the wrist-view is visualized for intermediate frames.

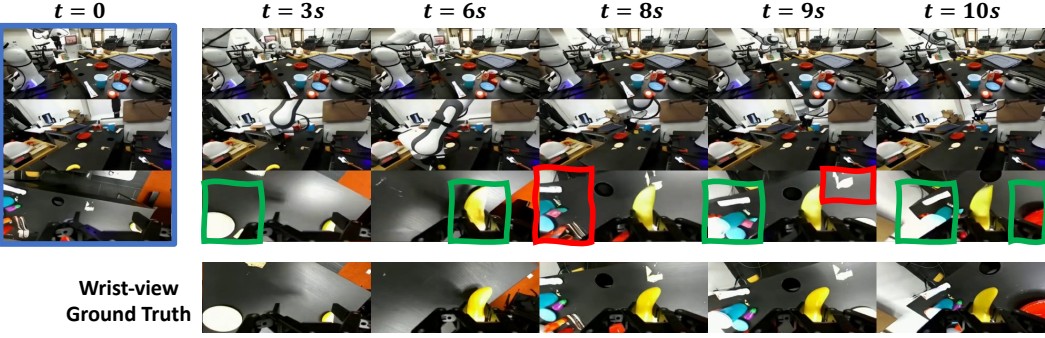

Figure 5: Consistency of Ctrl-World. Since the wrist camera's field of view changes dramatically within a single trajectory, leveraging multi-view information and memory retrieval is essential for generating consistent wrist-view predictions. Prediction highlighted in the green box are inferred from other camera views, while those in the red box are retrieved from memory.

methods WPE, IRASim and Ctrl-World-third-view all fail to move the green towel or the red bowl. In contrast, Ctrl-World precisely models the robot–object interactions through joint prediction of the wrist-camera view, which provides critical, fine-grained information about contact events and object state changes.

**Controllability of the World Model.** A key requirement of a world model is the ability to simulate diverse future outcomes conditioned on different actions. We find that our model exhibits fine-grained controllability, producing precise future predictions even for actions that differ by only a few centimeters (see Figure 4). We hypothesize that this controllability arises from two main factors: first, the dense action space coverage in the DROID dataset; and second, our use of multi-view prediction and frame-level action conditioning, which is also supported by our ablation studies. On the left side of Figure 4, we visualize the attention weights when predicting the $t = 4\,\mathrm{s}$ frame and observe strong attention to the $t = 0\,\mathrm{s}$ frame with a similar pose, highlighting the effectiveness of our memory retrieval mechanism.

**Consistency of the World Model.** For the wrist camera, since the camera's field of view changes dramatically within a single trajectory, it is challenging for models to generate consistent, long-term predictions. As shown in Figure 5, we find that our model effectively leverages relevant information

from both other camera views and historical frames, enabling it to generate consistent wrist-view predictions. Ablations on memory components and frame-level conditions are in Table 2, which confirm the importance of each component.

## 5.3 WORLD MODEL FOR POLICY EVALUATION

In this section, we evaluate whether Ctrl-World can be used to evaluate the instruction-following ability of generalist robot policies and accurately reflect their performance rankings in the real world (Li et al., 2024). We set up our own DROID platform and randomly place two third-person cameras around the workspace. Similar to how prior works have seen DROID policies generalize to new setups (Pertsch et al., 2025), we find that Ctrl-World, pretrained solely on the open-sourced DROID dataset, *can make accurate future predictions zero-shot in our newly configured scene with novel camera placements.*

**Policies and Tasks.** We evaluate three publicly released policies, $\pi_0$ (Black et al., 2023), $\pi_0$-FAST (Pertsch et al., 2025), and $\pi_{0.5}$ (Intelligence et al., 2025), across diverse tasks including Pick-and-Place, Towel-Folding, Drawer, Wipe-Table, Close-Laptop, Pull-tissue and Stack tasks on our DROID platform. We initialize real-world and world model rollouts with the same initial observations and execute each policy, following Algorithm 1. We report instruction following rates and success rates in Figure 7 and visualize qualitative comparisons between real and imagined rollouts in Figure 6. More rollout details can be found in Appendix B.

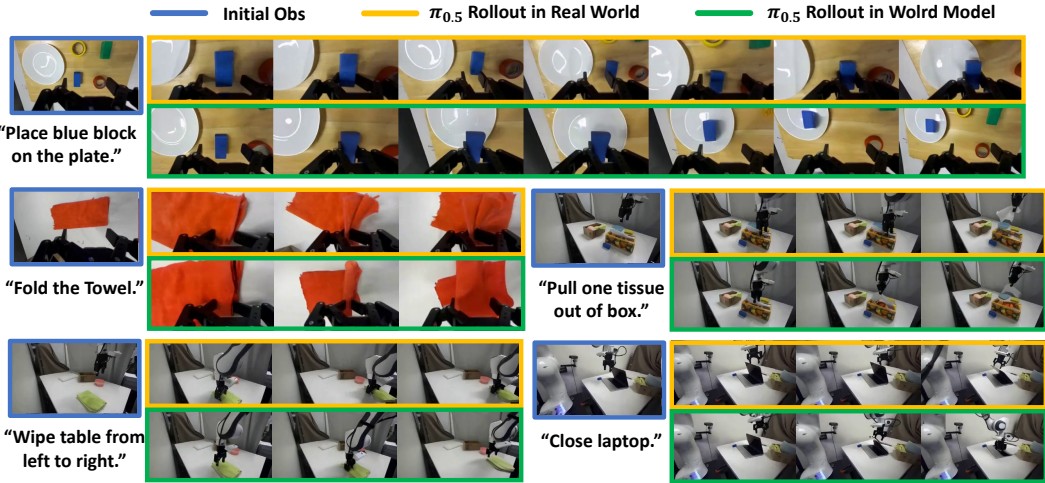

Figure 6: Comparisons between $\pi_{0.5}$ rollouts in the real-world and world model. Each trajectory contains 20 interactions between $\pi_{0.5}$ and Ctrl-World. Remarkably, both the generalist policy and Ctrl-World transfer zero-shot to our new DROID setup.

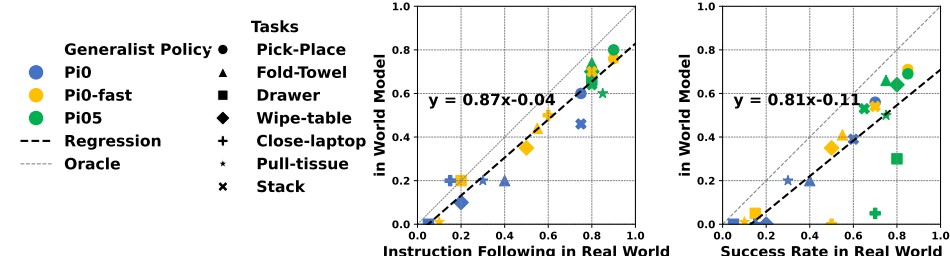

Figure 7: Quantitative correlations between real-world and world-model rollouts. The world model reliably captures instruction-following behavior but tends to underestimate the execution success rate.

**Comparison Between Real-World and World Model Rollouts.** Our results show that policy's high-level instruction-following behavior in the world model is closely correlated with that observed in the real world. However, we notice some gaps in evaluating low-level execution, specifically in

precise modeling of complex physics dynamics such as collisions, objects sliding away, rotations, etc. (e.g., interaction with laptop is imprecise in Figure 6). We also observe that generalist policies tend to keep retrying in the real world after failed attempts, which the world model sometimes does not capture. Although some failure trajectories are included in the DROID dataset, there are still many failure modes outside the data distribution. We expect that collecting additional in-domain policy rollout data would improve the fidelity of the learned dynamics and narrow this gap (Team, 2025).

## 5.4    WORLD MODEL FOR POLICY IMPROVEMENT

**Post-train Policy with Synthetic Data.** We now evaluate whether Ctrl-World can be used to generate synthetic post-training data for improving VLA models without real-world data. We use $\pi_{0.5}$ as our base policy and follow Algorithm 1. As described in Section 4.2, we encourage rollout diversity by either (1) rephrasing task instructions or (2) resetting the robot arm to a new initial state. For rephrasing, we call an LLM API (Team et al., 2023) to paraphrase instructions (e.g., transforming "place glove in box" into "pick up the cloth and put it inside the box"). For resetting, we randomly sample a new target initial position and move the robot arm there using a linear-interpolation motion planner before policy-interaction begins. We generate 400 trajectories per task and retain 25–50 successful trajectories based on human preference judgments. This selection step could be automated with reward models, which is an active area of research (Ma, 2025). Finally, we fine-tune the policy on the curated synthetic dataset for 2k steps, improving base model's capability in unfamiliar instructions and objects.

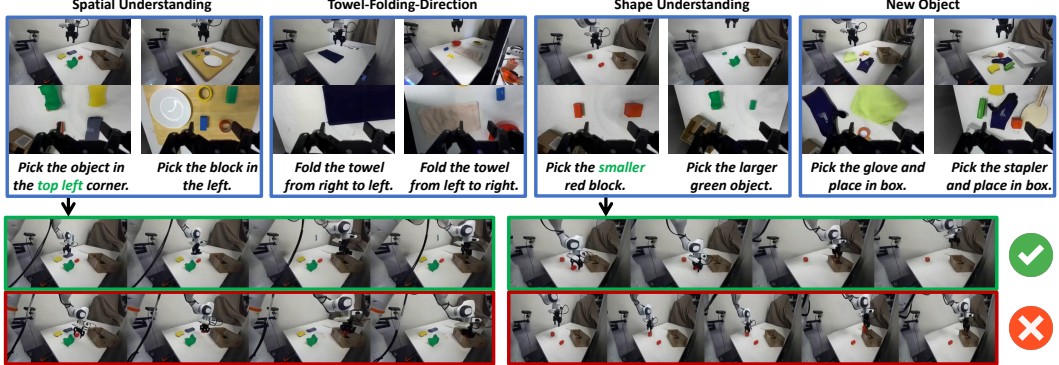

Figure 8: The top row illustrates examples of post-training tasks, while the bottom row presents synthetic trajectories generated within the world model. The world model can produce both successful and failed rollouts; we keep the successful trajectories and use them for policy fine-tuning.

**Results.** Some representative task examples and synthetic trajectories are visualized in Figure 8, and quantitative results are reported in Figure 9. While the pretrained $\pi_{0.5}$ policy achieves low success rates on unfamiliar objects and novel instructions, post-training aligns the model with new instructions and boosts the success rate from 38.7% to 83.4% on these downstream tasks. We include task details in Appendix C.

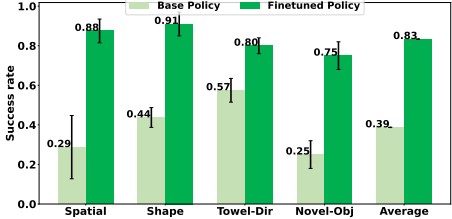

Figure 9: Policy improvement. Post-training on synthetic data improves policy instruction-following by 44.7% on average.

## 6    CONCLUSION

We presented Ctrl-World, a controllable world model for robot manipulation that supports closed-loop policy evaluation and improvement entirely within the model's imagination. Policies evaluated in Ctrl-World exhibit instruction-following behaviors that closely mirror those in the real world. Notably, post-training on generated data boosts the pretrained robot policy's success rate on novel instructions from 38.7% to 83.4%.

Despite these promising results, important challenges remain. Our model can fail on tasks involving precise interactions or long-horizon reasoning, and performance is sensitive to initial observations.

These limitations may diminish as video backbones become more physically accurate and coherent over time (Ball et al., 2025; Agarwal et al., 2025). In addition, our experiments focus on improving instruction following, and we expect that our model is not accurate enough to improve performance in other aspects such as the low-level success rate on previously seen instructions. Improving the model with iterative policy roll-out and fine-tuning is an exciting future direction. Looking forward, we believe generative world models can transform how robots acquire new skills, enabling scalable policy evaluation and allowing them to learn not just from real world experience, but also safely and efficiently from generated experience.

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

## A  MORE DETAILS FOR WORLD MODEL LEARNING

**Model Architecture.** Our world model closely follows the architecture of Stable Video Diffusion (SVD) (Blattmann et al., 2023a), and initializes from the SVD pretrained checkpoint. The only newly initialized component is a 3-layer MLP that projects 7-dimensional Cartesian-space actions into a 1024-dimensional latent embedding.

The input images are first encoded by a VAE with a spatial downsampling ratio of $8 \times 8$. In practice, we use $k = 7$ history frames, each perturbed with independent random noise to improve robustness. We set the action conditioning window to be one second, corresponding to 15 action steps. To reduce GPU memory consumption, we transform these 15 actions in the Cartesian space (see Section B) and temporally downsample them to 5 steps before feeding them into the model.

Each frame contains three $192 \times 320$ images, which are encoded into latent features of shape $24 \times 40$. The resulting total input token shape is $B \times (7 + 5) \times (3 \times 24 \times 40)$, which is then processed by the spatial-temporal transformer backbone.

**Training Datasets.** We use all 95k trajectories from the DROID dataset. For each training step, we randomly sample a trajectory and then uniformly sample a frame within that trajectory as the current frame. We then retrieve memory frames by sampling backward in time and set the model's prediction target to be the subsequent future frames.

**Training Process.** We train the model on 2×8 H100 GPUs with a total batch size of 64. The learning rate is set to be 1e-5, and we train for 100k steps, which takes approximately 2–3 days to complete.

## B  MORE DETAILS FOR POLICY EVALUATION

**Details on interaction between policy and world model.** We directly use the official $\pi_0$-DROID, $\pi_0$-FAST-DROID, and $\pi_{0.5}$-DROID policies from `https://github.com/Physical-Intelligence/openpi` to interact with Ctrl-World. To the best of our knowledge, Ctrl-World is the first world model that enables policy-in-the-loop interactions between state-of-the-art VLA model. These open-sourced policies take joint angles and two views of camera as input and output joint velocities. In contrast, our world model conditions on the end-effector pose in Cartesian space. To bridge this mismatch, we train an *adapter* on the DROID dataset that maps the current joint angles $q_t^{\text{joint}}$ and predicted joint velocities $a_{t+1:t+H}^{\text{jv}}$ into future joint configurations $q_{t+1:t+H}^{\text{joint}}$. We then apply Franka Panda forward kinematics (FK) to convert these joint configurations into Cartesian-space poses $q_{t+1:t+H}^{\text{cartesian}}$. The adapter is implemented as a simple two-layer MLP.

The overall process is as follows: given the current joint configuration $q_t^{\text{joint}}$, multi-view observation $o_t$, and language instruction $l$, the policy outputs $H$-step joint velocities:

$$a_{t+1:t+H}^{\text{jv}} = \pi(q_t^{\text{joint}}, o_t, l).$$

These are passed through the adapter to predict future joint configurations, followed by FK to compute Cartesian poses:

$$q_{t+1:t+H}^{\text{joint}} = \text{Adapter}(q_t^{\text{joint}}, a_{t+1:t+H}^{\text{jv}}), \qquad q_{t+1:t+H}^{\text{cartesian}} = FK(q_{t+1:t+H}^{\text{joint}}).$$

Finally, the world model predicts the next $H$ frames conditioned on the current observation, the calculated Cartesian poses, and the history Cartesian poses:

$$o_{t+1:t+H} = WM(o_t, q_{t+1:t+H}^{\text{cartesian}}, q_{\text{history}}^{\text{cartesian}}).$$

This setup enables fully autoregressive rollouts, allowing the official $\pi_0$-DROID, $\pi_0$-FAST-DROID, and $\pi_{0.5}$-DROID policies and Ctrl-World to interact seamlessly in imagination space.

| Task | Method | Instruction Following | | Success Rate | |
|------|--------|-----------|-------------|-----------|-------------|
| | | Real world | World Model | Real world | World Model |
| Pick-Place | $\pi_0$ | 0.75 | 0.60 | 0.70 | 0.55 |
| | $\pi_0$-fast | 0.90 | 0.75 | 0.85 | 0.70 |
| | $\pi_{0.5}$ | 0.90 | 0.80 | 0.85 | 0.70 |
| Fold-Towel | $\pi_0$ | 0.40 | 0.20 | 0.40 | 0.20 |
| | $\pi_0$-fast | 0.55 | 0.45 | 0.55 | 0.40 |
| | $\pi_{0.5}$ | 0.80 | 0.75 | 0.75 | 0.65 |
| Drawer | $\pi_0$ | 0.05 | 0.00 | 0.05 | 0.00 |
| | $\pi_0$-fast | 0.20 | 0.20 | 0.15 | 0.05 |
| | $\pi_{0.5}$ | 0.80 | 0.65 | 0.80 | 0.30 |
| Wipe-table | $\pi_0$ | 0.20 | 0.10 | 0.20 | 0.00 |
| | $\pi_0$-fast | 0.50 | 0.35 | 0.50 | 0.35 |
| | $\pi_{0.5}$ | 0.80 | 0.70 | 0.80 | 0.65 |
| Close-laptop | $\pi_0$ | 0.15 | 0.20 | 0.15 | 0.00 |
| | $\pi_0$-fast | 0.60 | 0.50 | 0.50 | 0.00 |
| | $\pi_{0.5}$ | 0.80 | 0.70 | 0.70 | 0.05 |
| Pull-tissue | $\pi_0$ | 0.30 | 0.20 | 0.30 | 0.20 |
| | $\pi_0$-fast | 0.10 | 0.0 | 0.10 | 0.0 |
| | $\pi_{0.5}$ | 0.85 | 0.60 | 0.75 | 0.50 |
| Stack | $\pi_0$ | 0.75 | 0.45 | 0.60 | 0.40 |
| | $\pi_0$-fast | 0.80 | 0.70 | 0.70 | 0.55 |
| | $\pi_{0.5}$ | 0.80 | 0.65 | 0.65 | 0.55 |

Table 3: Comparison of instruction-following and success rate across methods and tasks.

**Breakdown for policy evaluation.** We present the instruction-following and low-level execution success rates in Table 3.

**Task details and criterion.** In our experiments, we use human annotators to evaluate whether each trajectory is a success or a failure. Although this evaluation process can be automated in the future using large vision-language reward models, our focus in this paper is on the world model itself, so we rely on human preference as the reward signal. We provide clear criteria to determine whether a trajectory merely follows the instruction or achieves full task success:

- **Pick-place**: Several objects and receptacles are placed on the tabletop. The instruction is of the form "Pick up A and place in B." A trajectory is considered to follow the instruction if the policy attempts to grasp the correct object $A$. It is considered a success if object $A$ is successfully placed into the target receptacle $B$.

- **Fold the Towel:** A towel is lying flat on the table, with other objects possibly present. The instruction is "Fold the towel." A trajectory is considered to follow the instruction if the gripper moves to the towel's edge and attempts to lift and fold it. A trajectory is considered successful if the towel's surface area becomes half in the end.

- **Drawer:** The instruction is to "Place object A into drawer". A trajectory follows the instruction if the robot attempts to place object A inside the drawer. It is a success if object A is eventually placed in the drawer.

- **Wipe Table:** The instruction is to wipe the table surface. A trajectory follows the instruction if the gripper makes contact with the towel and moves in a sweeping motion. It is considered successful if a large portion of the table is covered by the sweeping motion.

- **Close Laptop:** The instruction is to close an open laptop. A trajectory follows the instruction if the gripper approaches the laptop lid. It is considered successful if the lid is fully closed.

- **Pull Tissue:** The instruction is to pull a tissue from a tissue box. A trajectory follows the instruction if the gripper approaches the tissue slot and pinches a tissue. It is considered successful if at least one tissue is fully extracted.

- **Stack:** The instruction is to stack one object on top of another. A trajectory follows the instruction if the gripper lifts the correct object. It is a success if the object is placed stably on top of the target object.

## C    MORE DETAILS FOR POLICY IMPROVEMENT

**Finetuning Process.**  We finetune $\pi_{0.5}$-DROID policy based on official codebase `https://github.com/Physical-Intelligence/openpi`. We finetune the pretrained checkpoint on our synthetic dataset for 2k steps on 4 H100 GPUs.

**Task Descriptions:**

- **Spatial Understanding Tasks:** 2–6 random objects are placed on the table. The policy is instructed to pick an object at a specified spatial location and place it in the box. Example instructions include: "Pick the object on the top-right side and place it in the box" or "Place the object on the far-left side into the box."
- **Shape Understanding Tasks:** 2–3 random objects are placed on the table, where some share the same attributes but differ in size. The policy must distinguish objects based on the size. Example instruction: "Pick the larger red block and place it in the box."
- **Towel-Folding with Directions:** A towel and other distractor are placed on the table, and the policy is given instructions specifying a particular folding direction (e.g., "Fold the towel from left to right").
- **Novel Objects:** We introduce unseen objects such as a glove and a stapler which Pretrained policy can not identify very well.

**Detailed success rate.**  We provide detailed task success rates inside each categories:

| | Left | Right | Bottom | Top | Left Top | Left Bottom | Right Top | Right Bottom | Average |
|---|---|---|---|---|---|---|---|---|---|
| Base Policy | 0.50 | 0.45 | 0.30 | 0.45 | 0.15 | 0.20 | 0.05 | 0.20 | 0.2875 |
| After Post-Training | 0.85 | 0.90 | 1.00 | 0.80 | 0.85 | 0.90 | 0.90 | 0.80 | 0.875 |

Table 4: Policy improvement (Spatial Understanding).

| | Big Left | Big Right | Small Left | Small Right | Average |
|---|---|---|---|---|---|
| Base Policy | 0.40 | 0.45 | 0.40 | 0.50 | 0.4374 |
| After Post-Training | 0.85 | 0.95 | 0.95 | 0.90 | 0.9125 |

Table 5: Policy improvement (Shape understanding).

| | Towel-1 | Towel-2 | Towel-3 | Towel-4 | Average |
|---|---|---|---|---|---|
| Base Policy | 0.60 | 0.50 | 0.55 | 0.65 | 0.575 |
| After Post-Training | 0.75 | 0.8 | 0.85 | 0.80 | 0.80 |

Table 6: Policy improvement (Towel folding with direction).

| | Novel-obj-glove | Novel-obj-stapler | Average |
|---|---|---|---|
| Base Policy | 0.20 | 0.30 | 0.25 |
| After Post-Training | 0.80 | 0.70 | 0.75 |

Table 7: Policy improvement (Novel object).

