# OpenReview forum: "Ctrl-World: A Controllable Generative World Model for Robot Manipulation"
_ICLR.cc/2026/Conference — ICLR 2026 Poster_

### Official Review · Reviewer_strQ · 2025-10-24

**Soundness:** 3
**Presentation:** 3
**Contribution:** 3
**Rating:** 6
**Confidence:** 5

**Summary:**

This paper proposes Ctrl-World, a controllable multi-view generative world model for robotic manipulation that enables policy-in-the-loop imagination rollouts for both policy evaluation and improvement. By integrating multi-view joint prediction, frame-level action conditioning, and pose-conditioned memory retrieval, the model achieves spatially consistent, controllable, and long-horizon video generation. Trained on the large-scale DROID dataset, Ctrl-World can simulate policy behavior across diverse camera views and environments, achieving close alignment with real-world performance and improving policy success rates by 44.7% through synthetic fine-tuning. Overall, the work offers a significant step toward scalable simulation-based evaluation and data-efficient improvement of generalist robot policies.

**Strengths:**

- This paper explores an interesting and important problem in the field of robotic manipulation, action-conditioned embodied video generation. Leveraging video generative models to simulate action policies represents a promising research direction, as it enables more efficient and scalable simulation, potentially facilitating the future development of RL in VLA systems.

- The proposed pipeline is well-designed and comprehensive. Through the incorporation of memory mechanisms, multi-view representations, and action-conditioning, it effectively ensures controllability between the generated videos and the corresponding actions.

- The paper also provides extensive real-world experiments. The authors validate the effectiveness of the proposed Ctrl-World framework in simulating policy models across different robotic platforms.

**Weaknesses:**

- The paper addresses an important problem in action-conditioned embodied video generation for robotic manipulation. However, I notice that similar approaches, such as EVAC [1], have also explored multi-view representations and sparse memory mechanisms. The paper would benefit from a more detailed discussion and comparison with such related work.

- Another concern lies in the simulation fidelity of object dynamics. For computational efficiency, the proposed method employs a relatively compact video diffusion network. It remains unclear whether the generated videos are sufficiently accurate for downstream simulation purposes—especially for deformable objects, where precise modeling of shape deformation is essential, and for articulated objects, where accurate representation of hinge or rotational motion is critical.

- Finally, the practical applicability of the proposed model requires further clarification. It would be valuable to discuss whether the generated data can be directly used for policy training or whether the model itself could function as a video-based simulator to enable scalable reinforcement learning. This aspect is crucial for assessing the long-term impact of the approach, but it is not thoroughly validated in the current version.

- A current limitation is that all demonstrated examples and evaluations focus exclusively on successful policy trajectories, lacking simulation or analysis of failure rollouts. This prevents verification of whether the generated videos truly reflect accurate action–visual mappings. The paper should further compare and analyze cases where, under identical initial states, both successful and failed action policies are executed to assess the model’s fidelity and controllability.

[1] EnerVerse-AC: Envisioning Embodied Environments with Action Condition. arxiv:2505

**Questions:**

This paper presents an interesting exploration of action-conditioned embodied video generation. My main questions are:

- How does the method differ from existing approaches such as EVAC, which also employ multi-view and sparse-memory designs?

- Can the model accurately generate videos for complex cases like deformable or articulated objects, given its lightweight diffusion architecture?

- Can the generated data be directly used for policy training, or can the model serve as a video-based simulator for scalable RL?

---

> ### Author Response · Authors · 2025-11-21
> **Response to Reviewer strQ - Thank you for reviewing our paper!**
>
> We sincerely appreciate your time and effort in reviewing our paper! We provide detailed discussions based on your review:
>
> **1. How does the method differ from EVAC [1]? (W1, Q1)**
>
> Thank you for pointing out this related work - we have cited it in the revised version. Our method differs from EVAC in several key aspects:
>
> 1. **Architecture:** EVAC converts actions into a special ray-direction map, which requires precise camera calibration. In contrast, we use a simple frame-level pose conditioning and do not require any camera calibration. We can even zero-shot generalize to new camera placements since the dataset has random camera views.
> 2. **Evaluation protocol:** We provide rigorous video quantitative evaluations (PSNR, FVD, rollout metrics) in addition to qualitative visualizations; EVAC reports no quantitative video-quality metrics.
> 3. **VLA policy improvement:** Although EVAC trains both single-view and multi-view models, it interacts with only one single-view policy. We verified world model can interact with advanced multi-view VLAs. Moreover, we explore using **synthetic rollouts to improve** VLA policies, which EVAC does not investigate.
>
> [1] EnerVerse-AC: Envisioning Embodied Environments with Action Condition. arxiv:2505
>
> ---
>
> **2. Simulation fidelity for deformable and articulated objects (W2, Q2)**
>
> A2: As shown in Section 5.3 (Figure 6), we evaluate many deformable-object tasks (e.g., towel folding, table wiping, tissue pulling) and articulated objects (e.g., laptops). All tasks are **zero-shot** transfers without any in-domain finetuning, so the predicted trajectories are not perfectly matched to real-world physics (but always reasonable). We believe post-training world model on in-domain data would further improve fidelity.
>
> ---
>
> **3. Application of the proposed model: can the generated data be directly used for policy training or scalable RL? (W3,Q3)**
>
> A3: Great question! In Section 5.4, we already filter high-quality generated trajectories and **directly finetune policies** on them, similar to the usage of synthetic data in the LLM domain [1]. We view our world model as a promising **neural simulator** that could enable scalable evaluation and reinforcement learning in the future.
>
> [1] Self-Instruct: Aligning Language Models with Self-Generated Instructions (ACL 2023)
>
> ---
>
> **4. Generating both success and failure trajectories from the same initial frame (W4)**
>
> A4: Yes — we include many such examples on our website. Ctrl-World is capable of producing both success and failure futures starting from the same initial observation: <https://sites.google.com/view/ctrl-world>
>
> ---
>
> Thank you again for reviewing our paper! We hope these additional experimental results can solve your concerns and we will include all of them in the new version of the paper. Any further questions are welcome!

---

### Official Review · Reviewer_bxDF · 2025-10-31

**Soundness:** 2
**Presentation:** 3
**Contribution:** 3
**Rating:** 6
**Confidence:** 4

**Summary:**

This paper proposes a method for building a multi-view world model that supports fine-grained action control and stable long-horizon generation. The authors conduct extensive experiments to verify the effectiveness of Ctrl-World in visual generation compared with baselines, and perform ablation studies to demonstrate the effectiveness of the three proposed techniques. In addition, the authors carry out policy evaluation and synthetic data experiments to validate the effectiveness of Ctrl-World for robot learning.

**Strengths:**

1. This paper addresses a practical problem and demonstrates the effectiveness of wrist-mounted cameras for robot world models.
2. The authors conduct detailed real-world experiments to verify the effectiveness of using world models for policy evaluation.
3. The synthetic data experiments help expand robotic arm datasets, and the experimental results in the paper show strong potential.

**Weaknesses:**

1. The technical novelty is relatively weak. Adding a wrist camera for joint prediction is not very challenging, and similar multi-view world model ideas already exist in autonomous driving, but are not discussed here [1].
2. The idea of Frame-level Action Conditioning (lines #185–#191) has already been used in [2][3], but here it is not properly cited or explained, which may confuse the source of the contribution.
3. Some doubts about Section 5.3: policy evaluation should test the same architecture with different checkpoints or slightly different training strategies. The world model should be able to reflect these differences; only then can it replace the real world to evaluate policies during development, instead of evaluating completely different architectures. Of course, testing on more architectures is good, but the key is whether the world model can detect subtle differences in success rates, similar to [4].
4. Some doubts about Section 5.4: the source of performance improvement is unclear — is it from rewritten instructions or from the world model? My main concern is reproducibility and real validity. This experiment is very similar to [5], but lacks detailed discussion and comparison.

[1] Cosmos-Drive-Dreams: Scalable Synthetic Driving Data Generation with World Foundation Models. Arxiv.

[2] IRASim: A Fine-Grained World Model for Robot Manipulation. ICCV 2025.

[3] Pre-trained video generative models as world simulators. Arxiv.

[4] SimplerEnv: Simulated Manipulation Policy Evaluation Environments for Real Robot Setups. CoRL 2024.

[5] DREAMGEN: Unlocking Generalization in Robot Learning through Video World Models. CoRL 2025.

**Questions:**

1. Some papers [1][2] discuss the issue of visual degradation, and their solution is to add small amounts of noise to the conditioning frames to mitigate distribution shift. What is the fundamental difference between that approach and the Pose-conditioned Memory Retrieval Mechanism proposed in this paper? Which method is more effective for addressing visual degradation?

2. The main difference from other action-conditioned world models is the addition of language instructions. This means video generation is jointly constrained by both the text and the action chunks. In this setting, what is the role of the text input? I am curious about the motivation for this design and whether there is strong evidence supporting its necessity.

[1] IRASim: A Fine-Grained World Model for Robot Manipulation. ICCV 2025.

[2] Training Agents Inside of Scalable World Models. Arxiv.

---

> ### Author Response · Authors · 2025-11-21
> **Response to Reviewer bxDF - Thank you for reviewing our paper!**
>
> We sincerely appreciate your time and efforts in reviewing our paper! We provide detailed discussions based on your review:
>
>
> **1. Adding noise to conditioning frame or pose-conditioned memory retrieval. Which one is more effective for visual degradation? (Q1)**
>
> **Our method actually uses both noisy conditioning and pose-conditioned memory retrieval.** Adding noise to conditioning frame makes the model robust to distribution shift. Pose-conditioned retrieval further allows the model to fetch relevant history frames with similar poses. Our ablations in Table 2 (Ctrl-World w/o frame-level pose-conditioning) confirm its importance: removing it significantly worsens both PSNR (23.56 → 21.2) and FVD (97.4 → 122.7).
>
> ---
>
> **2. The motivation for including language instructions in the world model (Q2)**
>
> SVD’s original architecture includes a CLIP image token, so we preserve this design to retain its pretrained semantic prior. This token mainly serves as a lightweight semantic info; in practice, we find the world model still follows the action conditioning faithfully.
>
> ---
>
>
> **3. Does improvement come from rewritten instructions or from the world model? (W4)**
>
> A3: The improvement comes from **high-quality synthetic rollout data**, similar to using synthetic data to improve LLM [1]. Rewriting instructions is just a way to get successful rollouts within the world model. We also use world-model resets in the paper.
> To clarify, we do ablations on different ways to collect synthetic data (rewrite-only / reset-only / both). We find that as long as we obtain enough successful synthetic rollouts, the final performance is similar.
>
> |Ablations on synthetic data   | Spatial Tasks |
> |---------------------------|--------|
> |Original policy | 0.29 |
> |ours+paraphrase | 0.80 |
> |ours+reset | 0.85 |
> |ours+both | **0.88** |
>
>
> [1] Self-Instruct: Aligning Language Models with Self-Generated Instructions (ACL 2023)
>
> ---
>
> **4. Test checkpoints with small differences (W3)**
>
>
>
> A4: We appreciate the reviewer’s suggestion. Unfortunately, for Pi0, Pi0-FAST, and Pi0.5 we only have access to a single released checkpoint per policy, so we cannot perform a controlled sweep over adjacent checkpoints. Our goal in Sec. 5.3 is correspondingly narrower: to test whether a single world model trained once on DROID can (i) generalize zero-shot to a new setup and (ii) recover the relative ranking across architecturally different generalist policies. As shown in Fig. 7, Ctrl-World achieves this, despite underestimating absolute success.
>
> We agree that distinguishing subtle differences between nearby checkpoints would be valuable. However, this is difficult even in real-robot evaluation for contact-rich tasks, where success rates are noisy and require many rollouts for statistically meaningful comparison [1,2]. Given remaining low-level inaccuracies in Ctrl-World (Sec. 5.3, Fig. 6–7), we do not expect it to resolve a few-percent difference between adjacent checkpoints. Our evaluation therefore focuses on comparing policies trained with different recipes, not on fine-grained hyperparameter tuning.
>
> We will clarify this scope in Sec. 5.3 and the Limitations section. We also agree that augmenting the world model with additional in-domain rollouts is a promising next step toward enabling checkpoint-level policy selection.
>
> [1] Yaxuan Li, Yichen Zhu, Junjie Wen, Chaomin Shen and Yi Xu. WorldEval: World Model as Real-World Robot Policies Evaluator (arXiv, May 2025)
>
> [2] Riccardo Mereu, Aidan Scannell, Yuxin Hou, Yi Zhao, Aditya Jitta, Antonio Dominguez, Luigi Acerbi, Amos Storkey, Paul Chang. Generative World Modelling for Humanoids: 1X World Model Challenge Technical Report (arXiv, Oct 2025)
>
> ---
>
> **5. Multi-view world model in autonomous driving domain (W1)**
>
> A5: Thank you for pointing this out — we have added the appropriate citations [1]. While multi-view generative modeling exists in autonomous driving, to the best of our knowledge, **Ctrl-World is novel in building such a controllable multi-view world model in the robot manipulation domain.**
>
> [1] Cosmos-Drive-Dreams: Scalable Synthetic Driving Data Generation with World Foundation Models. Arxiv.
>
> ---
>
> **6. Proper citation of frame-level action conditioning (W2)**
>
> A6: Thank you for the suggestion! We previously cited [2] in the related works section; we now explicitly cite [2,3] in the method section to clarify our contribution, and we additionally cite [4,5] in the evaluation section.
>
> ---
>
> Thank you again for reviewing our paper! We hope these additional experimental results can solve your concerns and we will include all of them in the new version of the paper. Any further questions are welcome!

---

### Official Review · Reviewer_gAW8 · 2025-11-02

**Soundness:** 2
**Presentation:** 3
**Contribution:** 2
**Rating:** 4
**Confidence:** 3

**Summary:**

The paper proposes CTRL-WORLD, a controllable, multi-view video-diffusion world model for robot manipulation designed to run policy-in-the-loop. Key components are: (i) joint multi-view prediction (including wrist views), (ii) frame-level action conditioning, and (iii) a pose-conditioned memory retrieval mechanism. The model is initialized from Stable Video Diffusion, fine-tuned on 95k trajectories, 564 scenes, and interacts with $\pi$0/$\pi$0-FAST/$\pi$0.5 policies via an adapter that maps joint-velocity actions to Cartesian poses. The authors report better video metrics than prior single-view world models, correlation between imagination-space and real-world rollouts, and a +44.7% improvement in instruction following for $\pi$0.5 after fine-tuning on curated synthetic rollouts.

**Strengths:**

1. The paper cleanly adapts a large video diffusion backbone into an action-conditioned, multi-view world model with explicit mechanisms for long-horizon consistency and control. The system is carefully described, including history frames, frame-wise cross-attention with poses/actions, and the memory stride idea.

2. The adapter that converts joint-velocity chunks into future Cartesian poses enables closed-loop interaction with modern VLA policies, which is practically useful.

3. Ablations show significant drops on both computation-based and model-based video metrics. Quantitative tables and qualitative rollouts are informative.

4. The paper demonstrates a concrete synthetic-data workflow, i.e., instruction paraphrasing with initial-state resets, as well as human-preference curation, that improves instruction following of $\pi$0.5 on downstream tasks.

**Weaknesses:**

1. Both the evaluation labels and the selection of synthetic trajectories rely on human preference judgments; the protocol, rater agreement, and quality controls are unspecified. Because only successful imaginations are kept for fine-tuning, there is a risk of optimistic bias in the reported improvement, especially absent size-matched controls versus random or uncurated synthetic data.

2. Policies act in joint-velocity space, whereas CTRL-WORLD conditions on Cartesian poses via an MLP + FK adapter. There is no error analysis or ablation isolating the adapter’s contribution. Given that several physics errors (e.g., collisions/sliding) are noted, the adapter may be a major bottleneck that is not rigorously evaluated.

3. The multi-view token concatenation, frame-level conditioning, and memory retrieval are each known ideas. The paper’s novelty is mainly their combination and scale on DROID. Stronger baselines, e.g., extending IRASim/WPE to multi-view, and theoretical insights are absent, the contribution feels incremental.

4. The text highlights generalization for “over 20 s”, but most quantitative evaluation reports 10 s trajectories. The horizon sensitivity is not analyzed, e.g., the memory stride/length hyperparameters.

**Questions:**

1. What happens if CTRL-WORLD conditions directly on joint-space action chunks, i.e., with or without poses? Please provide adapter error statistics and an end-to-end ablation isolating its effect on Fig. 7 correlations.

2. Can you extend IRASim/WPE to multi-view and re-run Table 1 to confirm that the gains are not primarily from adding wrist view rather than the proposed mechanisms?

3. Provide results varying rollout horizon (10–30 s) and memory parameters (k, m), showing how PSNR/FVD and instruction-following correlation degrade with horizon.

4. Please include size-matched controls: (a) random uncurated imaginations, (b) paraphrase-only vs. reset-only vs. both, (c) selection via a reward-model instead of humans, and (d) real DROID successes of equal count.

---

> ### Author Response · Authors · 2025-11-21
> **Response to Reviewer gAW8 - Thank you for reviewing our paper!**
>
> We sincerely appreciate your time and effort in reviewing our paper!
> Following your feedback, we conducted many additional experiments and we hope these new results and discussions will effectively address your concerns:
>
> ---
>
> **1. About baselines — extending IRASim/WPE to multi-view for stronger comparisons (W3, Q2)**
>
> Following your suggestion, we extended the official IRASim and WPE implementations to support multi-view prediction and report the results below.
> We find that directly adding multi-view to the baseline can not improve performance too much (even hurt performance for IRASim). We believe this is because predicting multi-view future is more challenging, especially with the wrist camera.
> These comparisons further highlight the effectiveness of Ctrl-World, which leverage memory-retrieval mechanisms and pretrained video model to improve multi-view performance.
>
> | | Third-view                     | PSNR ↑  | FVD ↓  | Wrist-view | PSNR ↑  | FVD ↓  |
> |---|---|---|---|---|---|---|
> | WPE-single-view           || 20.33 | 156.4        ||-|-|
> | IRASim-single-view        || **21.36** | 138.1      ||-|-|
> | Ours-single-view          || 21.27 | **127.5**       ||-|-|
> ||||||||
> | WPE-multiview (new)       || 21.17 |    147.1     | |15.26| 245.12 |
> | IRASim-multiview (new)    || 20.21 | 165.36   ||15.77|158.07|
> | Ours-multiview           || **23.56** | **97.4**       ||**19.18**|**127.1**|
>
>
> ---
>
> **2. Ablations on how to filter successful synthetic data: (a) uncurated rollout, (b) paraphrase-only vs. reset-only vs. both, (c) selection via a reward model, and (d) real DROID successes of equal count. (W1, Q4)**
>
> Thank you for the insightful comments! We conducted ablations across all four filtering strategies, keeping the size of the synthetic data fixed.
> (a) Fine-tuning the policy on uncurated synthetic rollouts does not improve performance, as the policy still imitates the original action distribution. (b) Paraphrase and reset have similar performance as long as they can generate the success data. (c)
> We find that strong VLMs (e.g., Gemini 2.5pro) are already highly reliable at judging trajectory success—close to human filtering quality. We added the same success criteria used by humans (Appendix B, line 839) directly into the VLM prompt. (d) Real-world rollouts naturally produce the highest-quality data but require substantially more rollout time.
>
>
> |Ablations on synthetic data   | Spatial Tasks |
> |----|-----|
> |Original policy | 0.29 |
> |(a)ours+uncurated | 0.26 |
> |(b)ours+paraphrase | 0.80 |
> |(b)ours+reset | 0.85 |
> |(c)ours+reward model | 0.80 |
> |(d)real robot(Oracle) | **0.93** |
> |ours+both | **0.88** |
>
>
> ---
>
> **3. About the action condition adapter: why not condition directly on joint-space action chunks? (W2, Q1)**
>
> We find that joint pose condition or Cartesian pose condition yields similar performance (they differ only by analytical FK). However, conditioning on joint velocities noticeably degrades performance. Our hypothesis is that joint velocity is much noisier and is a less stable conditioning signal for the world model.
>
> Moreover, different VLA policies operate in different action spaces (joint velocity, joint position, or end-effector pose). To ensure compatibility and strong performance across policies, we adopt Cartesian end-effector poses as the action condition in our paper.
>
> | Condition type      | Cartesian position | Joint position | Joint velocity |
> |--- |---|---|---|
> | PSNR ↑   | **23.56** | **23.20**        | 21.89|
> | FVD ↓   | **97.4** | **103.5**        | 120.2 |
>
> ---
>
>
> **4. Ablations on varying rollout horizon and memory parameters (k, m), why highlights “over 20s”. (W4, Q3)**
>
> We report video quality under rollout horizons of 5s, 10s, and 20s. We cap the horizon at 20s because most DROID trajectories do not extend beyond this length; this is also why the main paper emphasizes 20s and uses a 20-second rollout in the evaluation section 5.3. Increasing the rollout horizon naturally degrades PSNR/FVD due to error accumulation.
>
> | Rollout Horizon=      | 5s | 10s | 20s |
> |---- |-----|-------------|---|
> | PSNR ↑   | **24.44** | 23.56        | 22.65 |
> | FVD ↓   | **90.66** | 97.4       | 105.4 |
>
> We additionally sweep the memory stride m and history-frame count k. We find:
> (1) Longer effective history (m·k) improves PSNR because the model can leverage richer contextual cues.
> (2) An effective history of roughly 6 second yields the best FVD.
>
> | k=6 with m=     | 0.5s | 1s | 2s |
> |----------- |--------|-------------|---|
> | PSNR ↑   | 23.40 |   23.43     | **23.56** |
> | FVD ↓   | 98.6 |    **96.44**    | 97.4 |
>
> | m=2s with k=     | 1 | 3 | 6 |
> |----------- |--------|-------------|---|
> | PSNR ↑   | 23.06 |   23.35     | **23.56** |
> | FVD ↓   | 105.5 |    **95.6**     | 97.4 |
> ---
>
> Thank you again for reviewing our paper! We hope these additional experimental results can address your concerns and we will include all of them in the final version of the paper. Any further questions are welcome!

---

### Official Review · Reviewer_qxWr · 2025-11-02

**Soundness:** 4
**Presentation:** 4
**Contribution:** 3
**Rating:** 8
**Confidence:** 4

**Summary:**

This paper introduces a controllable world model designed for policy-in-the-loop learning, capable of generating multi-view, spatially, and temporally consistent rollouts conditioned on past robot states, observations, and future actions. The model is trained on the DROID dataset and validated in real-robot experiments. The proposed approach reportedly achieves substantial improvements in policy success rate through trajectory synthesis for supervised fine-tuning. Overall I think this is a solid paper with novel idea that using world-model-based data synthesis for policy fine-tuning, which holds strong potential for advancing applications of world models.

**Strengths:**

1. The idea of controllable world model with correct action conditioning and cross-frame memory retrieval is important for practical world model application, this paper provides a straightforward yet effective solution towards this goal.
2. The presentation of this paper is good. Sections are well-organized, results are delivered clearly with figures and plots. Illustrations are well-made and helpful.
3. The proposed world model can be taken as a "plug-in" strategy for various pre-trained policies, which may have wide applications.
4. The performance of the proposed world model looks great, the frame-level pose condition and memory retrieval mechanism seems effective.
5. The training cost of this world model is relatively small.

**Weaknesses:**

1. The proposed world model purely focus on the robot motion and takes interaction with other objects in the scene as environmental dynamics, which may straggle with tasks that involves multi objects or complex interactions.
2. Based on the videos, there are still gaps between real rollout and imagined rollout, especially when the end-effector interacts with objects.
3. The demonstrated tasks are still limited to "pick-and-place" style motions, it would be great to see failure cases on more complex robot tasks.

**Questions:**

1. It will be great to see failure cases for each experiments on the website.
2. Are the objects in the experiments ID or OOD? How does it generate to very novel objects?
3. What is the inference time?

---

> ### Author Response · Authors · 2025-11-21
> **Response to Reviewer qxWr - Thank you for reviewing our paper!**
>
> We sincerely appreciate your time and effort in reviewing our paper!
>
> ---
>
> **1. Are the objects in the experiments in-distribution or OOD? How does the model generalize to very novel objects?(Q2)**
>
> Thank you for the insightful question. Our model is trained on the open-sourced DROID dataset and deployed **zero-shot** in our new DROID setup, where the scenes, objects, and camera poses are all unseen during training. We believe the strong generalization ability comes from the diversity of the DROID dataset and the effectiveness of our model, enabling reasonable trajectories in novel environments.
>
> ---
>
> **2. It would be great to see failure cases(Q1)**
>
> Absolutely! We have included several failure cases on the project website — please feel free to check them out: <https://sites.google.com/view/ctrl-world>
>
> ---
>
> **3. About inference time(Q3)**
>
> Each interaction step runs 50 denoising iterations and takes approximately 4 seconds on a single H100 GPU. For each interaction cycle, we predict a 1-second future segment conditioned on the action chunk.
>
> ---
>
> **4. The proposed world model purely focuses on robot motion and treats interactions with objects as environmental dynamics. (W1)**
>
> Yes, that is correct. We condition the world model only on the robot-arm actions because the arm is the only **active degree of freedom**; object motions are passive and determined by the robot’s actions.
>
> ---
>
> **5. Gaps between real rollouts and imagined rollouts during interactions. (W2)**
>
> As mentioned in the limitations section, since we perform **no in-domain finetuning**, the model generally captures high-level behavior but can sometimes be inaccurate in the low-level physics, and the world model may deviate from the real trajectory during interactions. Post-training on in-domain data could be a promising future direction.
>
> ---
>
> **6. Tasks are limited to pick-and-place-style; more failure cases would be useful. (W3)**
>
> We actually include multiple deformable-object manipulation tasks—such as towel folding, table wiping, tissue, and sponge—which are difficult to simulate in traditional physics engines. Additional failure cases are available on the project website.
>
> ---
>
> Thank you again for your constructive comments and support. We hope our responses address your concerns, and we are happy to answer any further questions!

---

### Author Response · Authors · 2025-12-02
**General Response**

Dear ACs and Reviewers,

We sincerely thank the reviewers for their thoughtful feedback and constructive suggestions. We especially appreciate the recognition of the following aspects of our work:


## Review Highlights
- Motivation: Addressing a practical and important problem with strong application potential (**gAW8, bxDF, strQ**)
- Method: Simple yet well-designed method (**strQ, qxWr**) with concrete workflow (**gAW8**)
- Experiments: Strong model performance (**qxWr**), extensive real-world evaluations (**bxDF, strQ**), and useful ablation studies (**gAW8**)
- Presentation: Clear presentation (**qxWr**) and careful description (**gAW8**)

## Additional Experiments and Visualizations

(1) Reviewers **qxWr**, **bxDF**, and **strQ** provided positive initial scores (8, 6, and 6, respectively, with high confidence). We carefully addressed all their questions and added the missing citations as suggested.

In particular, Reviewers qxWr and strQ expressed interest in visualizations of failure cases with the same initial observation. We have added such examples to our project website (<https://sites.google.com/view/ctrl-world>), which better illustrate the capability of Ctrl-World.

(2) Reviewer **gAW8** gave an initial score of 4 and recommended adding more ablation studies and stronger baselines. Following this feedback, we conducted all the suggested experiments:
- We extend the baseline methods to the multi-view setting for a **stronger comparison**, which further validates the effectiveness of Ctrl-World.
- We **explore different strategies for filtering successful synthetic data**, further demonstrating the potential and robustness of our approach.
- We conduct additional ablation studies on our design choices, **including different action conditioning types and memory parameters**.

These additional experiments and stronger baselines further demonstrate the effectiveness and robustness of our approach.

We sincerely appreciate the reviewers’ insightful comments, which help strengthen this work. All the new results and revisions will be incorporated into the final version of the paper.

Sincerely,

The Authors

---

### Meta-Review · Area_Chair_vz8A · 2026-01-07

**Summary:**

The paper received broadly positive feedback from three of the four reviewers, who recognized its strong practical motivation.  Although one reviewer raised substantial concerns about incremental novelty and potential optimistic bias, the author provided a detailed response, but did not receive a response from the reviewer. Overall, the paper meets the standard of ICLR. But I hope the authors to carefully revise your paper with all issues addressed and responses in your final submission.

**Reviewer Concerns:**

The authors comprehensively responded to every weakness and question raised by all reviewers.

**Reviewer Scores:**

Reviewer qxWr (initial 8) would maintain their score as their requests for failure cases and object generalization were fully satisfied.
Reviewer bxDF (initial 6) would likely maintain or raise their score given the new multi-view baselines, citation updates, and clarification on instruction rewriting versus world model contribution.
Reviewer strQ (initial 6) would likely maintain or raise their score following the detailed comparison with EVAC, inclusion of failure rollouts, and discussion of deformable object fidelity.
Reviewer gAW8 (initial 4) did not respond to the author's rebuttal, and the low score may be retained.

---

### Decision · Program_Chairs · 2026-01-26

Accept (Poster)